DATA RELEASE

# Mycobacterial metabolic model development for drug target identification

Bridget P. Bannerman[1,2,*], Alexandru Oarga[3] and Jorge Júlvez[3]

1 Lucy Cavendish College, University of Cambridge, Lady Margaret Rd, Cambridge, CB3 0BU, UK
2 Science Resources Foundation, 128 City Road, London, EC1V 2NX, UK
3 Department of Computer Science and Systems Engineering, University of Zaragoza, C/María de Luna n° 1, 50018, Zaragoza, Spain

## ABSTRACT

Antibiotic resistance is increasing at an alarming rate, and three related mycobacteria are sources of widespread infections in humans. According to the World Health Organization, *Mycobacterium leprae*, which causes leprosy, is still endemic in tropical countries; *Mycobacterium tuberculosis* is the second leading infectious killer worldwide after COVID-19; and *Mycobacteroides abscessus*, a group of non-tuberculous mycobacteria, causes lung infections and other healthcare-associated infections in humans. Due to the rise in resistance to common antibacterial drugs, it is critical that we develop alternatives to traditional treatment procedures. Furthermore, an understanding of the biochemical mechanisms underlying pathogenic evolution is important for the treatment and management of these diseases.

In this study, metabolic models have been developed for two bacterial pathogens, *M. leprae* and *My. abscessus*, and a new computational tool has been used to identify potential drug targets, which are referred to as bottleneck reactions. The genes, reactions, and pathways in each of these organisms have been highlighted; the potential drug targets can be further explored as broad-spectrum antibacterials and the unique drug targets for each pathogen are significant for precision medicine initiatives. The models and associated datasets described in this paper are available in GigaDB, Biomodels, and PatMeDB repositories.

**Subjects**  Microbiology, Applied Microbiology, Medical Microbiology, Systems Biology

**Submitted:**   12 November 2022

* Corresponding author. E-mail: bpc28@cam.ac.uk

Preprint submitted at https://doi.org/10.1101/2023.03.31.534705

## DATA DESCRIPTION

### Context

*Mycobacterium leprae* (NCBI:txid1769) and *Mycobacterium tuberculosis* (NCBI:txid1773) are two related pathogenic mycobacteria responsible for leprosy and tuberculosis in humans. Another related mycobacterium, *Mycobacteroides abscessus* (NCBI:txid36809), causes opportunistic infections in healthcare-related settings. Previous analyses of the metabolic models of *M. tuberculosis* have supported studies demonstrating the evolutionary drivers of antibiotic resistance and the identification of novel drug targets against mycobacteria [1, 2]. Here, we demonstrate newly built genome-scale metabolic models of *M. leprae* and *My. abscessus*, including curation, simulation, and model-optimisation strategies [3–5]. To ensure the development of standardised metabolic models for the global systems biology community, we have implemented the recently released community standards and used Metabolic Model Testing MEMOTE quality control software to evaluate our models [6, 7].

**Figure 1.** Mycobacterial metabolic model development for drug target identification: *My. abscessus* and *M. leprae*.

## METHODS

## Genome-scale metabolic model reconstruction, curation, and simulation

Automated draft reconstructions of *M. leprae* and *My. abscessus* were downloaded from BioModels and evaluated against other organism-specific databases, such as BioCyc, (RRID:SCR_002298) and Kyoto Encyclopedia of Genes and Genomes (KEGG, RRID:SCR_012773) [8, 9]. COBRApy (RRID:SCR_012096), a Python toolbox for the construction, manipulation, and analysis of constraint-based models [3], and GNU Linear Programming Kit (GLPK: RRID:SCR_012764), a software package that solves efficiently large-scale linear programming problems [10], were used to manipulate and simulate the models.

To create new genome-scale metabolic models (GEMs) of *My. abscessus* and *M. leprae*, additional reactions, gene-to-reaction associations, and pathways (that were not in the automated model) were integrated from KEGG and BioCyc [8, 9]. The annotations of genes and metabolites were improved by comparing and transferring annotations from the related *M. tuberculosis* models in the iEKVIII model and BioCyc database [1, 8]. Further improvements to the models were made by comparing them with the compound formula and charge from the MetaNetX database, and by mapping the genes and reactions of the GEMs to the BiGG (RRID:SCR_005809), ChEBI (RRID:SCR_002088), KEGG (RRID:SCR_012773), and MetaCyc databases (RRID:SCR_007778) [11–13]. Figure 1 describes the full process of the mycobacterial metabolic models (*i*Mab22 and *i*Mlep22) of the pathogens *My. abscessus* and *M. leprae*. The revised model reconstructions of *M. leprae* (*i*Mlep22) and *My. abscessus* (*i*Mab22) can be instantiated without error on the COBRA software (version 0.16.0) [3].

### Biomass reactions for *M. leprae* and *My. abscessus*

We generated biomass reactions for *My. abscessu*s and *M. leprae* using the methodology in the pathway tools software [14] and the corresponding BioCyc database [15]. The software



tool *findCPcli*, which implements the computational method for identifying bottleneck reactions as drug targets, is available in GitHub [16].

## DATA VALIDATION AND QUALITY CONTROL

### Model optimisation

The dead-end metabolite reactions that were previously present in the automated model were eliminated to enhance the models' quality. The models were then iteratively evaluated, considering model-specific reactions for the *My. abscessus* and *M. leprae* organisms and comparisons with the BioCyc, KEGG, MetaNetX 4.2, BiGG, and ChEBI databases [8, 11–13].

To improve the quality of the models, the reactions with dead-end metabolites previously found in the automated models were removed. MEMOTE, a standardised genome-scale metabolic model testing programme, was used to undertake quality control checks during the models' iterations and optimisation [7]. In the process, Systems Biology Ontology (SBO) annotations and gene annotations from the KEGG database and 728 new formulae from the MetaNetX database were added [13]. As a result, the MEMOTE score increased from 49% to 63% on the *M. leprae* model and from 48% to 66% on the *My. abscessus* model.

The new GEMs for *M. abscessus* and *M. leprae* are encoded in the Systems Biology Markup Language (SBML) [17] and designated as follows:

- *i*Mlep22 for the *M. leprae* model: i for *in silico*, Mlep for *M. leprae*, and published in 2022. *i*Mlep22 consists of 5,625 reactions, 4,016 metabolites, and 871 genes.
- *i*Mab22 for the *My. abscessus* model: i for *in silico*, Mab for *My. abscessus*, and published in 2022. *i*Mab22 consists of 8,580 reactions, 6,273 metabolites, and 1,837 genes.

Standardisation and curation have been done according to the community standards for the development of metabolic models, as described in Carey *et al.* and Lieven *et al.* [6, 7], to produce gold-standard metabolic network reconstructions of *My. abscessus* and *M. leprae*. The development of *My. abscessus* and *M. leprae* (*i*Mab22 and *i*Mlep22) metabolic models is illustrated in Figure 1, and the overall capability of the models has been summarised in Figure 2.

### Comparative analysis

*My. abscessus*, an opportunistic pathogen, has a large genome size of 5.1 MB [18], with more biochemical and bottleneck reactions in the *i*Mab22 model compared with the obligate pathogens *M. tuberculosis* (with a genome size of 4.4 MB) and *M. leprae* (with an even smaller genome size of 3.3 bp) [19]. The fewer metabolic and bottleneck reactions in the *i*Mlep22 model can be attributed to the reductive evolution that occurred in *M. leprae* and the subsequent loss of genes. An illustration of the distribution of unique enzymes and bottleneck reactions in each of the models (*i*Mab22 and *i*Mlep22) in comparison with each other and with *M. tuberculosis* is demonstrated in Figure 3. Alternative enzymes are not included in this analysis because they catalyse the same biochemical reactions and do not fit into the category of bottleneck reactions, which are defined as unique reactions responsible for the survival and growth of the organisms in the metabolic network [4, 5, 20].

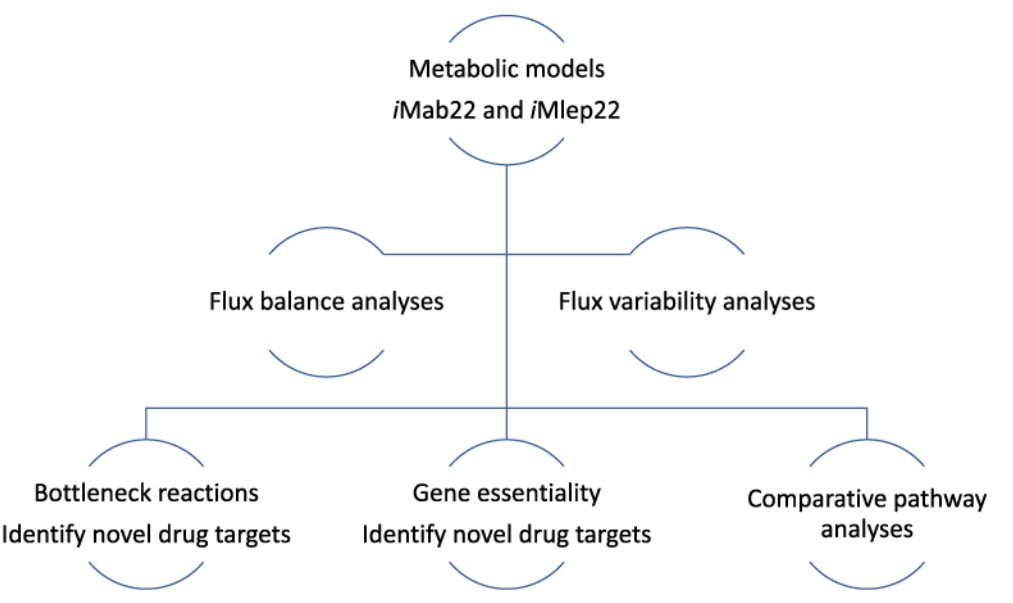

**Figure 2.** Overall capability of the models: *My. abscessus* (*i*Mab22) and *M. leprae* and (*i*Mlep22).

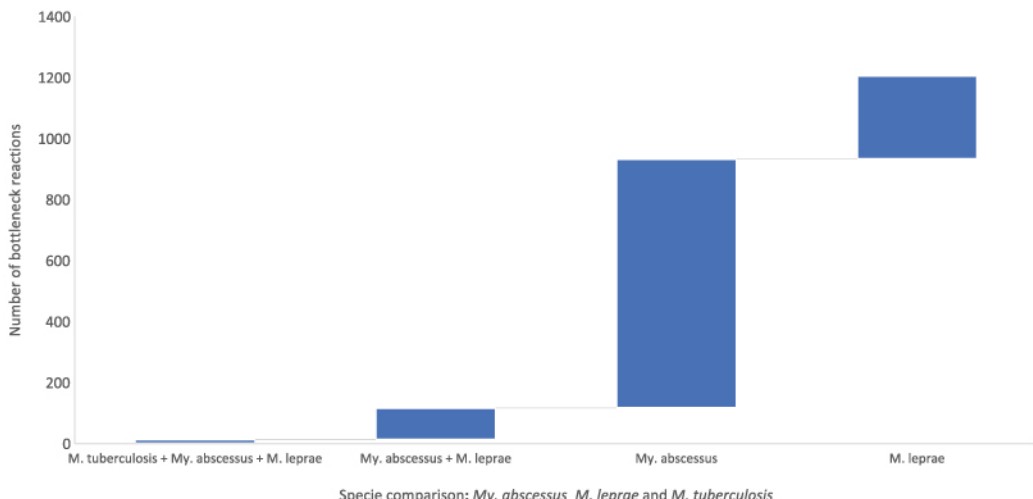

**Figure 3.** Unique/bottleneck reactions in *My. abscessus* and *M. leprae* in relation to *M. tuberculosis*.

## REUSE POTENTIAL

The standardised genomic scale metabolic models for *M. leprae* (*i*Mlep22) and *My. abscessus* (*i*Mab22) have been developed using the systems biology community standards for quality control and evaluation of models [6, 7], and are available for reuse by the global scientific community.

## DATA AVAILABILITY

Data supporting this work are openly available in the GigaDB repository [21]. The models can be retrieved from:



*My. abscessus* https://www.ebi.ac.uk/biomodels/MODEL2203300002 and https://www.patmedb.org/Bacteria/Mabscessus.

*M. leprae* https://www.ebi.ac.uk/biomodels/MODEL2203300001 and https://www.patmedb.org/Bacteria/Mleprae.

The *findCPcli* tool (RRID:SCR_023391, biotools:findcpcli) [4] used for analysing the models can be retrieved from GitHub [16].

## DECLARATIONS

### List of abbreviations

GEM: Genome-scale metabolic model; GLPK: GNU Linear Programming Kit; KEGG: Kyoto Encyclopedia of Genes and Genomes; MEMOTE: Metabolic Model Testing; CObraPy: COnstraints-based reconstruction and analysis for Python; SBO: Systems Biology Ontology.

### Ethics approval and consent to participate

The authors declare that ethical approval was not required for this type of research.

### Competing Interests

The authors declare no competing interests.

### Authors' contributions

BPB: conceptualization, data curation, software, formal analysis, investigation, methodology, and writing—original draft, review, and editing.

JJ: software, formal analysis and writing—review, and editing.

AO: data curation, software, formal analysis, and writing—original draft, review, and editing.

### Funding

The authors of this manuscript are supported by the Spanish Ministry of Science, Innovation, and Universities (JJ and AO).

### Acknowledgements

The authors would like to thank Ebirien Nte and Jo Chukualim for the graphic designs and scripting.

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
