## [Reviewer Report]

Reviewer name and names of any other individual's who aided in reviewer Grace MugumbateDo you understand and agree to our policy of having open and named reviews, and having your review included with the published papers. (If no, please inform the editor that you cannot review this manuscript.)YesIs the language of sufficient quality?YesPlease add additional comments on language quality to clarify if needed
First person reporting has been used with the word "We' used extensively. Are all data available and do they match the descriptions in the paper? YesAdditional CommentsAre the data and metadata consistent with relevant minimum information or reporting standards? See GigaDB checklists for examples <a href="http://gigadb.org/site/guide" target="_blank">http://gigadb.org/site/guide</a>NoAdditional CommentsThere is need to specify the type, size, standardisation and curation of the data that was used, especially when additional data was obtained from different databases.Is the data acquisition clear, complete and methodologically sound?YesAdditional CommentsSources of data are indicated in the paper, however the size of the data sets and type of data is not clear. Is there sufficient detail in the methods and data-processing steps to allow reproduction?NoAdditional CommentsThere is need to give more detail in the methods for reproducibility.Is there sufficient data validation and statistical analyses of data quality? NoAdditional CommentsValidation was performed, however no statistical analyses was mentioned. Is the validation suitable for this type of data?YesAdditional CommentsIs there sufficient information for others to reuse this dataset or integrate it with other data?NoAdditional CommentsMore detail is needed on data retrieval to allow reuse of the dataset.Any Additional Overall Comments to the AuthorThe Authors presented their work entitled 'Mycobacterial Metabolic Model Development for Drug Target Identification'. This is very innovative work that led to generation of M. laprae and M. abscessus models, important tools for drug target identification. Target identification for a number of infectious diseases provides information for structure-based molecular modification of new and alternative diseases. The target specific compounds will help reduce side effects among other things. Generation of the models by the authors is commendable.  There are a few corrections: 1) Under Abstract: Line 4: Please note that Mycobacterium tuberculosis is not a disease but the bacterium that causes the diseases tuberculosis. 2) Mehtods, GEM reconstruction, curation and simulation (i) Line two: Name the "other organism specific databases" (ii) Give a brief description of the COBRApy and the GLPK even if the source had been given. 3) The Method section need to be more informative to allow for reproducibility.RecommendationMinor Revision

---

## [Reviewer Report]

Reviewer name and names of any other individual's who aided in reviewer Nagasuma ChandraDo you understand and agree to our policy of having open and named reviews, and having your review included with the published papers. (If no, please inform the editor that you cannot review this manuscript.)YesIs the language of sufficient quality?YesPlease add additional comments on language quality to clarify if needed
Are all data available and do they match the descriptions in the paper? YesAdditional CommentsAre the data and metadata consistent with relevant minimum information or reporting standards? See GigaDB checklists for examples <a href="http://gigadb.org/site/guide" target="_blank">http://gigadb.org/site/guide</a>YesAdditional CommentsIs the data acquisition clear, complete and methodologically sound?YesAdditional CommentsIs there sufficient detail in the methods and data-processing steps to allow reproduction?YesAdditional CommentsIt would be useful if the authors could comment on how the models vary between the two species and with respect to M. tuberculosis. Specifically, a note on how the authors deal with alternate enzymes and whether they included enzymes specific to each species, would be helpful. Is there sufficient data validation and statistical analyses of data quality? YesAdditional CommentsIs the validation suitable for this type of data?YesAdditional CommentsA figure depicting the overall capability of the models would be usefulIs there sufficient information for others to reuse this dataset or integrate it with other data?YesAdditional CommentsAny Additional Overall Comments to the AuthorGenome-scale metabolic models are useful to the community as they can be used to address a variety of questions. It would be useful if the authors could include a section on the comparative performance of the models and link it to the known metabolic capability of these microbes.RecommendationMinor Revision